# Diagnostic Value of Superb Microvascular Imaging in Differentiating Benign and Malignant Breast Tumors: A Systematic Review and Meta-Analysis

**DOI:** 10.3390/diagnostics12112648

**Published:** 2022-10-31

**Authors:** Jiaping Feng, Jianghao Lu, Chunchun Jin, Yihao Chen, Sihan Chen, Guoqiang Guo, Xuehao Gong

**Affiliations:** 1Graduate School, Guangzhou Medical University, Guangzhou 510180, China; 2Department of Ultrasound, Shenzhen Second People’s Hospital, The First Affiliated Hospital of Shenzhen University, Sungang West Road 3002, Futian District, Shenzhen 518025, China

**Keywords:** breast neoplasms, superb microvascular imaging, diagnosis, meta-analysis, ultrasonography

## Abstract

Purpose: We performed a systematic review and meta-analysis of studies that investigated the diagnostic performance of Superb Microvascular Imaging (SMI) in differentiating between benign and malignant breast tumors. Methods: Studies published between January 2010 and March 2022 were retrieved by online literature search conducted in PubMed, Embase, Cochrane Library, Web of Science, China Biology Medicine Disc, China National Knowledge Infrastructure, Wanfang, and Vip databases. Pooled sensitivity, specificity, and diagnostic odd ratios were calculated using Stata software 15.0. Heterogeneity among the included studies was assessed using *I*^2^ statistic and Q test. Meta-regression and subgroup analyses were conducted to investigate potential sources of heterogeneity. Influence analysis was conducted to determine the robustness of the pooled conclusions. Deeks’ funnel plot asymmetry test was performed to assess publication bias. A summary receiver operating characteristic curve (SROC) was constructed. Results: Twenty-three studies involving 2749 breast lesions were included in our meta-analysis. The pooled sensitivity and specificity were 0.80 (95% confidence interval [CI], 0.77–0.84, inconsistency index [*I*^2^] = 28.32%) and 0.84 (95% CI, 0.79–0.88, *I*^2^ = 89.36%), respectively. The pooled diagnostic odds ratio was 19.95 (95% CI, 14.84–26.82). The area under the SROC (AUC) was 0.85 (95% CI, 0.81–0.87). Conclusion: SMI has a relatively high sensitivity, specificity, and accuracy for differentiating between benign and malignant breast lesions. It represents a promising supplementary technique for the diagnosis of breast neoplasms.

## 1. Introduction

Breast cancer, the most common neoplasm in women, is the leading cause of cancer-related female deaths worldwide. According to the Global Cancer Statistics (2020), breast cancer accounted for 11.7% of all new cancer diagnoses and 6.9% of all cancer-related deaths [1]. Early diagnosis is crucial for the prognosis of breast cancer patients. Wulburga et al. [2] demonstrated that early diagnosis of breast cancer is associated with a distinctly higher 5-year survival probability than diagnosis at a late stage (62.5% versus 35.8%). In addition, tumor neovascularization is strongly associated with tumor growth, invasion, and metastasis. Previous research has demonstrated that tumor growth stage can be divided into “pre-angiogenesis stage” and “angiogenesis stage”, according to whether the vascular diameter is >2 mm. When breast tumor is in the pre-angiogenesis stage (<2 mm in diameter), the extent of infiltrates is very limited; however, when tumor progresses to the angiogenesis stage (>2 mm in diameter), there is proliferation of blood vessels within lesion resulting in capillary infiltration into the surrounding tissue to absorb nutrients for sustaining tumor invasive growth and metastasis [3]. Therefore, early detection and characterization of tumor neovascularization are of great significance for the diagnosis and treatment of breast cancer [4]. Immunohistochemical evaluation of microvascular density (MVD) is the gold standard for evaluating tumor angiogenesis. However, the measurement of MVD is invasive and only available after surgery, which limits its application [5].

As a noninvasive, cost-effective, and radiation-free option, conventional ultrasound is an important screening modality for the early detection of breast lesions [6]. In addition, Color Doppler Flowing Imaging (CDFI) and Power Doppler Imaging (PDI) are widely used to display the number and characteristics of blood vessels within breast lesions [7]. However, several studies suggested that it may be difficult for CDFI and PDI to differentiate microvessels (<0.1 mm in diameter) within breast lesions [8,9,10].

With the rapid development of computer techniques in clinical ultrasonography [11], Superb Microvascular Imaging (SMI), a novel doppler technique, has been increasingly used in recent years. Moreover, the imaging principles of SMI are significantly different from CDFI. CDFI entails application of a single dimensional wall filter to remove the artifact, which contributes to the loss of low-speed flow components. In contrast, SMI uses multi-dimensional filter to separate flow signals from clutter, thus removing only the motion artifact and preserving the slow flow signals [12]. Currently, SMI is widely used for imaging of liver, thyroid, breast, and cervical lymph nodes [13].

A meta-analysis addressing this topic was published, which included 15 studies with 2071 breast lesions examined using SMI [14]. However, it had some limitations. First, the inclusion and exclusion criteria for studies were relatively less rigorous. For example, the purpose of a study included in their review was to investigate the effectiveness of SMI for the differentiation of intraductal breast lesions, not the overall breast lesions [15]. the results of the meta-analysis may have been affected by some data extraction errors [8,16,17]. Lastly, although they performed meta-regression and subgroup analysis to investigate the sources of heterogeneity, the setting of heterogeneity factors is relatively unreasonable.

Given these limitations, there was insufficient data to assess the accuracy of SMI for differentiation of breast lesions. It is necessary to reappraise the diagnostic value of SMI in differentiating benign and malignant breast tumors. In addition, there are four major diagnostic criteria of breast tumors in SMI, namely: Alder classification (AC), the presence of penetrating vessel (PV), microvascular distribution pattern (MVDP), and vascular index (VI). However, the diagnostic performance of SMI in previous studies showed variability due to the use of different diagnostic criteria. Therefore, the aim of our systematic review was to evaluate the diagnostic accuracy of SMI in differentiating benign and malignant breast tumors and to identify the diagnostic criteria associated with the best diagnostic performance.

Generally, this systematic review is structured as follows: “Introduction”, “Materials and Methods”, “Results”, “Discussion”, and” Conclusions”.

## 2. Materials and Methods

The guidelines of Preferred Reporting Items for Systematic Reviews and Meta-Analyses (PRISMA) of Diagnostic Test Accuracy (DTA) were followed in this systematic review [18]. which was registered on PROSPERO (https://www.crd.york.ac.uk/prospero/ (accessed on 1 February 2022)) (*registration number: CRD42021262596). In addition, the organization of the article is inspired by some high-quality studies [19,20].

### 2.1. Literature Search Strategy

Relevant studies published between January 2010 and March 2022 were identified by online literature search conducted in PubMed, Embase, Cochrane Library, Web of Science, China Biology Medicine Disc, China National Knowledge Infrastructure, Wanfang, and Vip databases. Medical Subject Headings (MeSH) terms as well as free words were adopted. The following search terms were used in our search strategy: “breast neoplasm (or breast carcinoma, breast cancer, breast tumor, mammary carcinoma, mammary cancer, mammary tumor)”, “superb microvascular imaging (or microvascular imaging, SMI),” “ultrasonography (or ultrasound, sonography)”, and “sensitivity and specificity (or predictive value of tests, predictive value, accuracy)”. In addition, the reference lists of relevant articles were manually screened to identify additional studies. 

### 2.2. Study Selection

The EndNote software 20.0 (Clarivate Analytics, Philadelphia, PA, USA) was used to manage the literature retrieved from electronic databases. Two reviewers independently screened the retrieved studies for eligibility. The inclusion criteria were: (1) original research articles in which patients with suspected breast lesions were screened using SMI (regardless of whether they were assessed with other imaging modalities, such as conventional ultrasonography, mammography, CEUS, SWE, or MRI); (2) at least one of the four diagnostic criteria mentioned above was used; (3) true-positive (TP), false-positive (FP), false-negative (FN), and true negative (TN) rates could be extracted directly or indirectly; (4) 50 or more lesions were included in the study; (5) language of publication: English and Chinese; and (6) the nature of breast tumors was confirmed by surgery or aspiration biopsy. 

The exclusion criteria were: (1) review articles, case reports, meta-analyses, letters, or conference abstracts; (2) duplicated publications; (3) the studies restricted to specific breast neoplasms, such as intraductal papilloma or small breast malignant tumor (less than 10 mm in diameter); and (4) the outcome indicator (TP, FP, FN, TN) cannot be directly or indirectly extracted from the article. Conflicts between individual judgments were resolved by discussion or consultation with a third author. 

### 2.3. Data Extraction 

The 2 reviewers independently extracted data pertaining to the following variables using Microsoft Excel spreadsheets: (1) general study characteristics including the first author’s name, country, publication year, number of lesions, mean age of patients, mean size of lesions, instrument used for SMI, scale, study design, diagnostic criteria; and (2) The TP, FP, FN, and TN were obtained directly or calculated based on the reported sensitivity and specificity values. In case of any missing information, the original authors were contacted for the relevant information. 

### 2.4. Quality Assessment 

All the included studies were independently assessed by 2 reviewers using the Quality Assessment of Diagnostic Accuracy Studies checklist version 2 (QUADAS-2). QUADAS-2 includes 4 items in the bias risk domain and 3 items in the applicability domain. The 4 items in the bias risk domain are: (1) patient selection; (2) index test; (3) reference standard; and (4) flow and timing. The 3 items in the applicability domain are: (1) patient selection; (2) index test; and (3) reference standard. Each item within the domains of QUADAS-2 was classified as low, high, and unclear risk. If any of the items in the domain were judged as no, it was judged as high risk. Only if all items in the domain were judged as yes, it was judged as low risk. If one or more of the items in the domain were judged to be uncertain, it was judged as uncertain risk. The quality assessment graphs were prepared using RevMan software 5.3 (The Nordic Cochrane Center, The Cochrane Collaboration, 2014).

### 2.5. Strategy for Data Synthesis

This study was conducted based on the guidelines for systematic reviews and meta-analyses [21,22]. TPs, FPs, FNs and TNs were used to calculate the pooled sensitivity, specificity, positive- and negative-likelihood ratios (PLRs and NLRs), and the diagnostic odds ratios (DORs) with corresponding 95% confidence intervals (CI) [23]. In addition, the summary receiver operating characteristic (SROC) analysis was performed to determine the diagnostic accuracy by measuring the area under the curve [24]. 

The Q test and the *I*^2^ statistic were used to assess the heterogeneity of sensitivity and specificity among studies. *p* values < 0.1 for Q test or *I*^2^ values > 0.5 were considered indicative of significant heterogeneity [25]. By sequentially removing one study at a time, influence analysis was conducted to determine the robustness of the pooled conclusions. Further, to investigate the source of heterogeneity, univariate meta-regression and subgroup analysis were conducted. Lastly, publication bias was evaluated with the Deeks’ funnel plot asymmetry test [26]. All statistical analyses were performed using Stata Version 15.0 (Stata Corp, College Station, TX, USA). 

### 2.6. Heterogeneity Assessment 

Some studies have shown that the epidemiology, risk factors, and tumor characteristics of breast cancer in China differ from those in Korea and other Asian countries [27,28,29]. In addition, VI is a quantitative diagnostic criteria with low operator dependence, whereas AC, PV and MVDP are qualitative diagnostic criteria with relatively high operator dependence [30]. Based on the country, diagnosis criteria, and study design, meta-regression and subgroup analysis were conducted to investigate the sources of heterogeneity. These definitions and grouping methods for included studies are described below in detail. If the country was China, it was marked Yes, otherwise marked No. Moreover, if the diagnostic criteria were VI, it was marked Yes, otherwise marked No. Furthermore, if the diagnosis basis was a prospective design, it was marked Yes, otherwise marked No. As for the definition of subgroup, the included study population was divided into two groups based on China or non-China: groups 1 and 2; diagnostic criteria were divided into four groups based on AC, PV, MVDP, and VI: groups 1,2,3 and 4. The included study design was divided into two groups based on study design: groups 1 and 2.

## 3. Results

### 3.1. Study Selection 

A total of 404 articles were identified after the initial search; of these, 184 duplicated studies were removed. After reviewing the titles and abstracts, another 82 articles were excluded. After full-text assessment of the literature, 26 articles were considered eligible based on the inclusion and exclusion criteria. It is worth mentioning that two types of diagnostic basis were respectively used in two eligible studies. Ultimately, 28 studies were included in the qualitative review. In addition, a further five studies were excluded due to incomplete data. Hence, 23 studies involving 2749 lesions were included in the meta-analysis. A schematic illustration of the literature screening and selection process is presented in Figure 1 (checklist seen in Appendix A).

### 3.2. Study Characteristics

The included studies were published between 2015 and 2021. Overall, 3732 lesions in 3579 patients were analyzed. The average age of the patients was 48.7 (range, 43.7–54.2) and the mean number of lesions per study was 133 (range, 50–300). Majority of the included studies were conducted in China [16,17,31,32,33,34,35,36,37,38,39,40,41,42,43,44,45,46,47,48]; five studies were conducted in Korea [30,49,50,51] and two studies were conducted in Turkey [52,53]. In all 23 studies, Alder classification, the presence of penetrating vessel, microvascular distribution pattern and VI were used as the diagnostic criteria of breast neoplasms and pathological results were used as the reference standard. Other main characteristics, including equipment used for SMI, scale and study design are presented in Table 1.

### 3.3. Quality Assessment 

The results of quality assessment are presented in Figure 2 and Figure 3. With respect to the risk of bias, three studies were judged to have high risk of bias in the “patient selection” domain for not enrolling a consecutive and random sample of patients. Moreover, four studies were judged to have unclear risk of bias in the “patient selection” domain due to lack of information on whether a consecutive or random sample of patients was enrolled.11 studies had unclear risk of bias in the “index test” domain, since it was not clear whether a threshold was pre-specified, in addition, ten studies were judged to have unclear risk of bias due to lack of information on whether the index test results were interpreted without knowledge of the reference standard. As for “reference standard” domain, due to lack of information on whether the reference standard was interpreted without knowledge of the index test results, 14 studies were found to have unclear risk of bias. Moreover, there was a lack of information to determine whether reference standards correctly classified target conditions in three studies.

As for “flow and timing” domain, the analysis for all included patients was not performed in one study, leading to a high risk of bias. In addition, 22 studies were judged to have unclear risk of bias since an appropriate interval between index test and reference standard was not elaborated. 

With respect to concerns regarding applicability, there were five studies with unclear concerns regarding patient selection, 14 studies with unclear concerns regarding index tests, and six with unclear concerns regarding references standard. In general, the quality of the included studies was relatively satisfactory.

### 3.4. Quantitative Synthesis

#### 3.4.1. Diagnostic Accuracy Assessment

As shown in Figure 4, the pooled sensitivity and specificity were 0.80 (95% CI, 0.77–0.84) and 0.84 (95% CI, 0.79–0.88), respectively. There was slight heterogeneity in sensitivity (*I*^2^ = 28.32%, *p* = 0.10) and significant heterogeneity in specificity (*I*^2^ = 89.36%, *p* < 0.01). SROC for the diagnostic value of SMI using VI for differentiation of benign and malignant breast lesions is shown in Figure 5. The area under SROC (AUC) was 0.85. Figure 6 shows the summary of positive-likelihood ratios (PLRs) and negative-likelihood ratios (NLRs) of SMI for differentiating between benign and malignant breast lesions. The pooled PLR and NLR of all eligible studies were 4.88 (95% CI, 3.77–6.31) and 0.24 (95% CI, 0.22–0.27), respectively. In terms of NLR, there was a slight heterogeneity (*I*^2^ = 17.87%, *p* = 0.22) and in terms of PLR, there was a significant heterogeneity (*I*^2^ = 76.23%, *p* < 0.01). The pooled diagnostic odds ratio was 19.95 (95% CI, 14.84–26.82) (Figure 7). Moreover, the result of influence analysis showed the relative stability of the pooled conclusion of our study. (Figure 8).

#### 3.4.2. Assessment of Publication Bias

Publication bias was assessed using the Deeks’ funnel plot asymmetry test. The results indicated no significant influence of publication bias of eligible studies (Figure 9).

### 3.5. Heterogeneity Assessment

We conducted meta-regression and subgroup analysis to investigate the potential source of heterogeneity. According to Figure 10, there was no statistically significant difference between sensitivity and specificity for patient mean age and lesion mean size. However, there is significant difference in sensitivity for the region of patients (*p* < 0.001). In addition, there are significant difference in both sensitivity and specificity for prospective study design and VI diagnostic (*p* < 0.001). A summary of the sensitivities, specificities, PLRs, NLRs and DORs of studies is presented in Table 2. The pooled specificities, PLRs values and DORs of the China group were higher than those of the non-China group. The pooled specificity, PLR values and DOR of the MVDP were higher than the other diagnostic criteria groups.

## 4. Discussion

### 4.1. Principal Results

In our current study, we evaluated the diagnostic performance of SMI in differentiating between benign and malignant breast lesions. In the 23 included studies, the sensitivity ranged from 0.77 to 0.82 and specificity ranged from 0.79 to 0.88. The pooled sensitivity and specificity values were 0.80 and 0.83, respectively. The pooled DOR was 19.95 and overall diagnostic accuracy, represented by AUC, was 85%. In addition, the pooled PLR and NLR of all eligible studies were 4.88 and 0.24, respectively, which indicated that SMI can be used to distinguish between benign and malignant breast tumors. However, due to the significant heterogeneity of specificity and PLR, we conducted a meta-regression and subgroup analysis. Meta-regression of the included studies indicated that region, study design, and diagnostic criteria might be potential sources of heterogeneity. However, these results require verification with follow-up studies due to the limited sample in this study. Subgroup analysis was also carried out for the diagnostic performance of SMI to differentiate benign and malignant breast lesions. There were higher pooled specificity, PLRs, and DORs in the China group compared to non-China group. It is worth mentioning that the pooled specificity, PLRs values, and DORs of the MVDP were higher than those of the other diagnostic criteria groups. Moreover, VI was roughly as effective as AC when used as the diagnostic basis for breast lesions on SMI. It is possible that the plane outlined by the sonographer with the richest blood flow signals is not the plane with the most obvious malignant characteristics, which may lead to missing important diagnostic information of breast lesions [50].

### 4.2. Comparison with Previous Systematic Review

A similar study was published, and the results supported the effectiveness of SMI in diagnosing breast lesions [14]. However, the meta-analysis had less rigorous criteria for selection of studies and data extraction, which may affect the credibility of the results. We conducted a relatively comprehensive literature search and employed more rigorous study selection criteria. Moreover, the application of QUADAS-2 tool is another strength of our work. Furthermore, we conducted more rational subgroup analyses to identify the sources of heterogeneity. For example, publication language should not be considered as a source of heterogeneity, because a study can be published in any language and the region of study may actually be the source of heterogeneity. Therefore, the previous study did not identify the source of heterogeneity, whereas our study suggests that region, study design, and diagnostic criteria may be potential sources of heterogeneity.

### 4.3. Clinical Implications of our Findings

SMI was shown to be superior to CDFI and PDI in detecting microvascular blood flow signals owing to its intelligent adaptive algorithm and powerful multidimensional wall filtering system [12]. At present, there are four main diagnostic criteria for breast tumors based on SMI, namely: AC, PV, MVDP, and VI. However, the choice of diagnostic basis is a key challenge for sonographers in clinical practice.

AC and VI are used to evaluate blood flow abundance of breast lesions. AC can be classified as grade 0, I, II, and III based on the number of blood vessels in the tumor [54]. However, the judgment of Alder grade may be affected by operator experience [51]. VI is defined as the ratio of blood vessels pixels within a lesion to pixels throughout the entire lesion, which can be calculated automatically by the built-in software of the instruments [4]. Lee et al. [30] demonstrated excellent intra-observer reproducibility and inter-observer reproducibility with respect to VI. In our meta-analysis, the pooled diagnostic performance of VI was equal to that of AC. Therefore, for inexperienced sonographers in clinical practice, VI may be a better choice, since it has similar diagnostic performance to AC but is more reproducible.

SMI enables visualization of breast tumor microvascular architecture. SMI mainly evaluates the microvascular structure from two perspectives: PVs and MVDP. In the subgroup analyses, the pooled diagnostic accuracy of the MVDP was higher than that of PV. Hence, the combination of VI and MVDP may unlock the potential of SMI for differentiating breast lesions. However, there is a paucity of relevant studies, and this requires validation in the future.

### 4.4. Limitations of Our Work

Some limitations of our study should be acknowledged. First, all eligible studies were conducted only in Asia, which may limit the generalizability of our results to patient populations in other parts of the world. Second, the parameter setting, and image acquisition details were not available for all studies, such as frame rate, dynamic range, and image acquisition section. It is possible that these factors contributed to heterogeneity of our study. Lastly, there were 23 eligible studies, however, only 2 of these studies used PVs as a diagnostic criterion, so a larger sample size may provide more convincing evidence. Further comprehensive studies are required to resolve these issues. 

## 5. Conclusions

Our systematic review suggests that SMI has an encouraging diagnostic value to differentiate between benign and malignant breast tumors. Moreover, according to the subgroup analyses, MVDP may be more effective among the four main diagnostic criteria, and it may benefit sonographers in selecting appropriate diagnostic criteria. In addition, study design, region, and the selection of diagnostic criteria may be potential sources of heterogeneity. In spite of its current limitations, such as a restricted region and a relatively small sample size, SMI remains a promising supplementary tool for sonographers to distinguish between breast lesions in clinical practice.

## Figures and Tables

**Figure 1 diagnostics-12-02648-f001:**
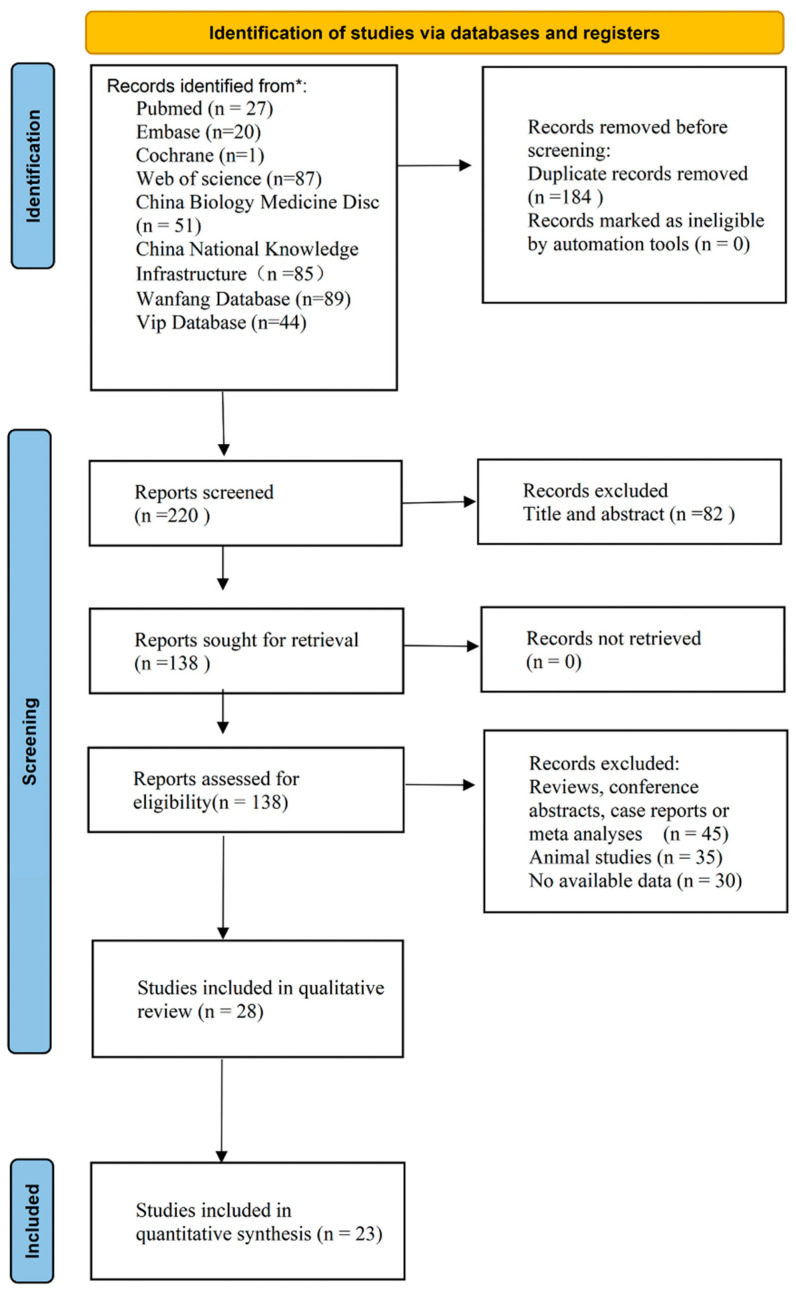
PRISMA flow diagram depicting the literature search and study selection. *: Both Chinese and English databases have been systematically searched.

**Figure 2 diagnostics-12-02648-f002:**
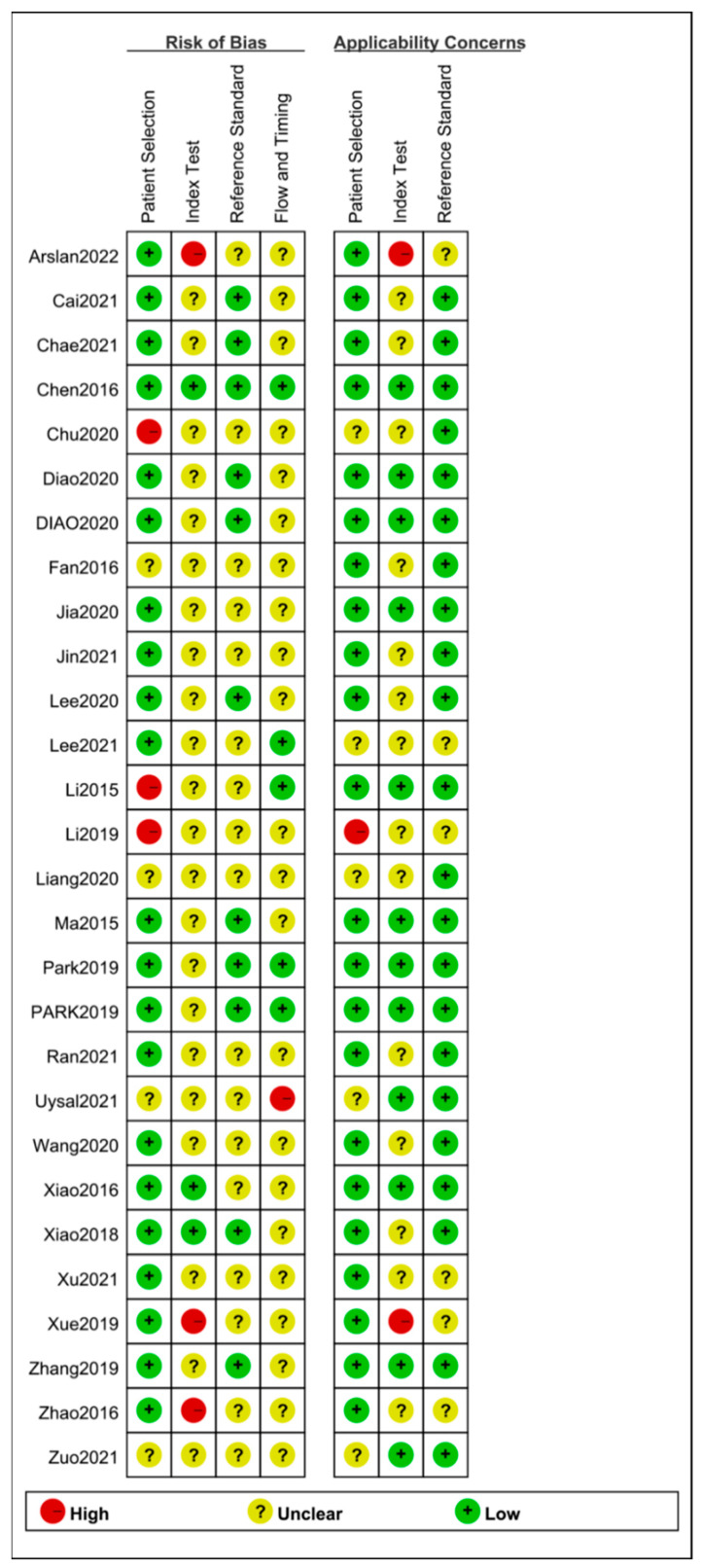
Risk of bias and applicability concerns summary: review authors’ judgements about each domain for each eligible study.

**Figure 3 diagnostics-12-02648-f003:**
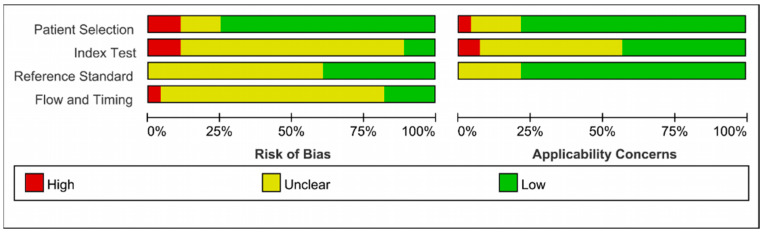
Risk of bias and applicability concerns graph: review authors’ judgements about each domain presented as percentages across eligible studies.

**Figure 4 diagnostics-12-02648-f004:**
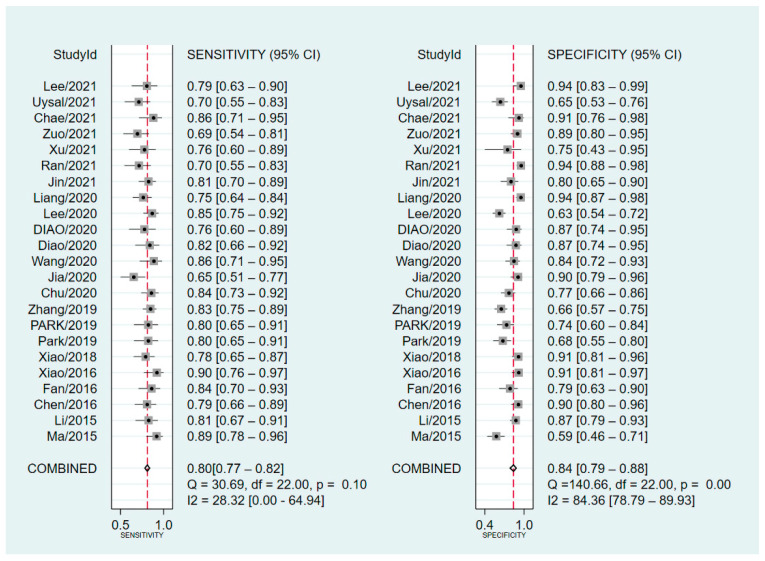
Pooled sensitivity and specificity for Superb Microvascular Imaging to differentiate benign and malignant breast lesions.

**Figure 5 diagnostics-12-02648-f005:**
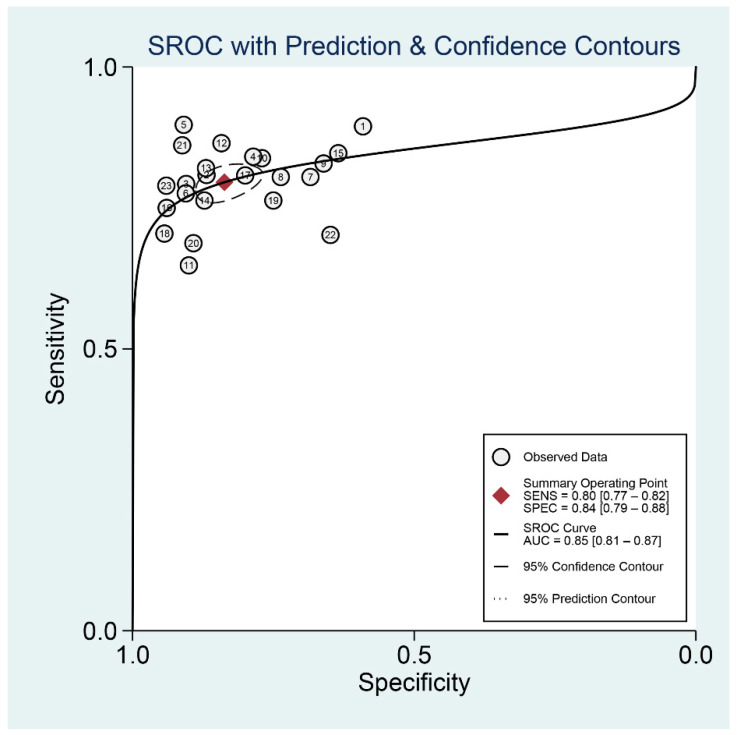
Summary receiver operating characteristic model for Superb Microvascular Imaging to differentiate benign and malignant breast lesions.

**Figure 6 diagnostics-12-02648-f006:**
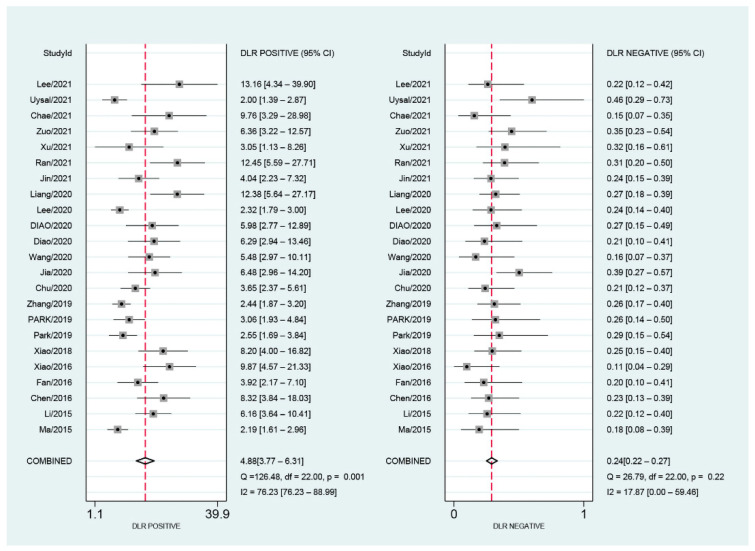
Pooled positive and negative likelihood ratios for Superb Microvascular Imaging to differentiate benign and malignant breast lesions.

**Figure 7 diagnostics-12-02648-f007:**
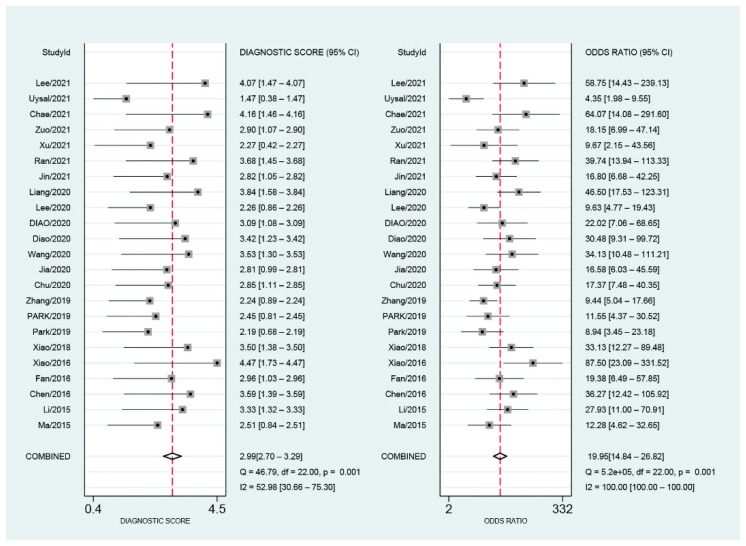
Pooled diagnostic odds ratio for Superb Microvascular Imaging to differentiate benign and malignant breast lesions.

**Figure 8 diagnostics-12-02648-f008:**
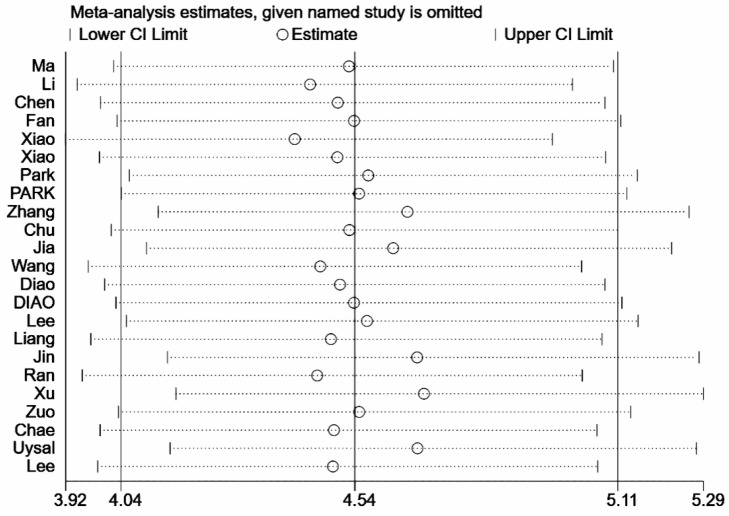
Influence analysis for eligible studies.

**Figure 9 diagnostics-12-02648-f009:**
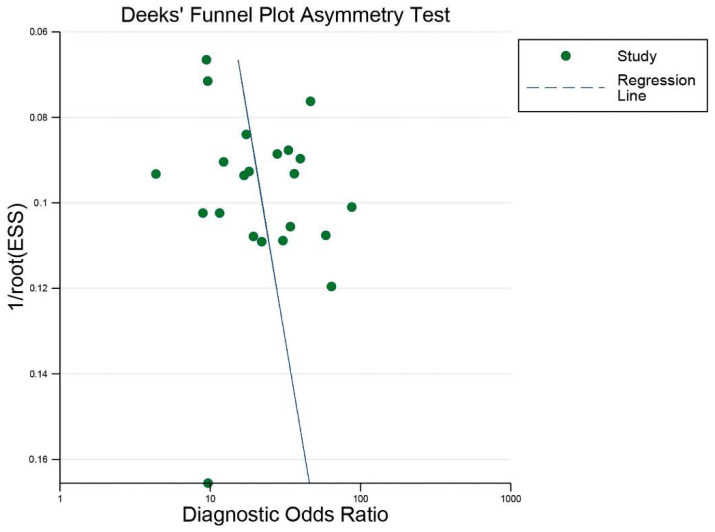
Deeks’ funnel plot of publication bias. No potential publication bias for all eligible studies was observed (*p* = 0.26). ESS: effective sample size.

**Figure 10 diagnostics-12-02648-f010:**
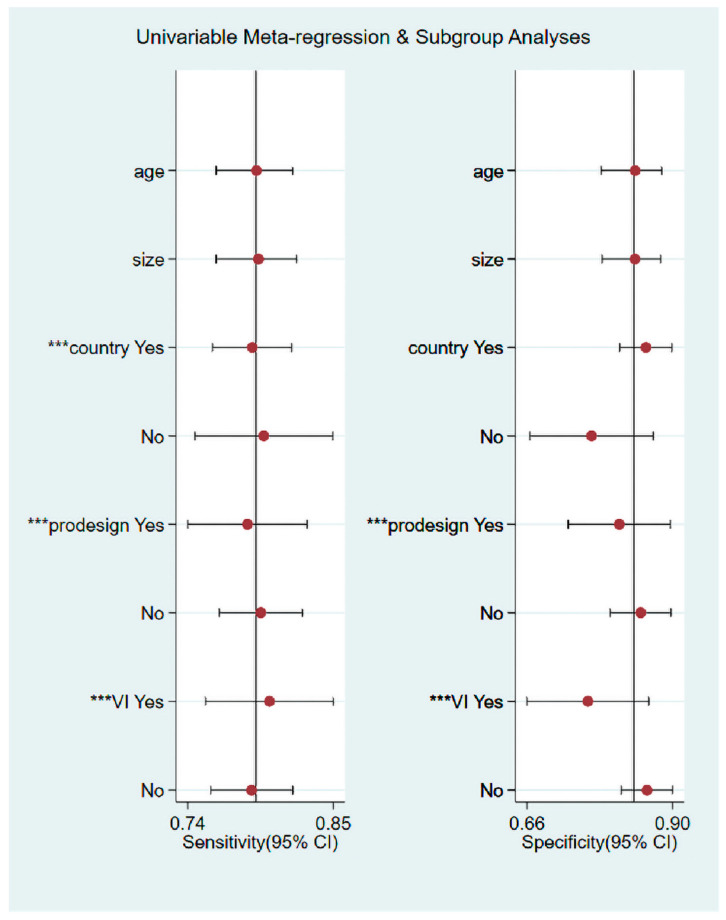
Meta-regression of included studies showing that patient country, predesign and VI as potential sources of heterogeneity. ***: *p* < 0.001.

**Table 1 diagnostics-12-02648-t001:** Baseline characteristics of identified studies and enrolled patients.

Author	Year	Country	Age (Mean, Year)	Size (Mean, mm)	Instrument	Scale (cm/s)	lesions	SD	DC	TP	FP	TN	FN
Ma	2015	China	43.7	22.4	Aplio 400	1–2	123	Pro	AC	51	27	39	6
Li	2015	China	44.1	18.4	Aplio 500	2.6	146	Retro	AC	38	13	86	9
Chen	2016	China	44.9	17.7	Aplio 400	1–2	116	Retro	MVDP	42	6	57	11
Fan	2016	China	47.5	17.5	Aplio 500	1.2	86	Retro	AC	37	9	33	7
Zhao	2016	China	44.5	14.9	Aplio 500	1.5–2.5	135	Retro	NA	NA	NA	NA	NA
Xiao	2016	China	44.1	18.3	Aplio 400	NA	132	Retro	MVDP	45	7	67	13
Xiao	2018	China	47.2	19	Aplio 500	1–2	105	Retro	MVDP	35	6	60	4
Park	2019	Korea	45.6	18.7	Aplio 500	3	98	Pro	PV	33	18	39	8
PARK	2019	Korea	45.6	18.7	Aplio 500	3	98	Pro	VI	33	15	42	8
Zhang	2019	China	44.9	23.5	Aplio 500	1.2–1.6	236	Pro	VI	92	39	76	19
Xue	2019	China	NA	NA	Aplio 500	NA	300	Retro	NA	NA	NA	NA	NA
Chu	2020	China	44.4	19.9	Aplio 500	1.2	142	Retro	AC	57	17	57	11
Jia	2020	China	48	22.4	Aplio 900	NA	114	Retro	MVDP	35	6	54	19
Wang	2020	China	49.9	29.7	Aplio 500	1.2	94	Retro	AC	32	9	48	5
Diao	2020	China	54.2	15.3	Aplio 500	2	85	Retro	PV	32	6	40	7
DIAO	2020	China	54.2	15.3	Aplio 500	2	85	Retro	MVDP	29	6	41	9
Lee	2020	Korea	49	11.3	Aplio 800	2.5	200	Retro	VI	72	42	73	13
Liang	2020	China	43.9	17.4	Aplio 400	1.2	177	Pro	MVDP	57	6	93	19
Li	2020	China	49.8	NA	Aplio 500	NA	208	Retro	NA	NA	NA	NA	NA
Jin	2021	China	NA	NA	Aplio 500	NA	123	Retro	AC	63	9	36	15
Ran	2021	China	49.4	20.1	Aplio 500	1.2	150	Pro	AC	31	6	100	13
Xu	2021	China	53.1	27.7	Aplio 800	NA	50	Retro	AC	29	3	9	9
Zuo	2021	China	NA	16	Aplio 400	NA	122	Pro	MVDP	33	8	66	15
Chae	2021	Korea	54.1	NA	Aplio 800	2.5	70	Retro	VI	31	3	31	5
Uysal	2021	Turkey	50.5	NA	Aplio 500	3.5	121	Retro	VI	33	26	48	14
Cai	2021	China	46.1	23.5	Aplio 500	1–2	238	Retro	VI	NA	NA	NA	NA
Lee	2021	Korea	46	10.7	Aplio 800	2.5	88	Retro	VI	30	3	47	8
Aralan	2022	Turkey	49	21.9	Aplio 300	1.5–2.5	90	Retro	VI	NA	NA	NA	NA

AC, Alder classification; DC, diagnostic criteria; FN, False negative; FP, false positive; NA, not available; PV, penetrating vessel; MVDP, microvascular distribution pattern; SD, study design; TN, true negative; TP, true positive.

**Table 2 diagnostics-12-02648-t002:** Subgroup analyses for diagnostic values of Superb Microvascular Imaging to differentiate benign and malignant breast lesions.

Subgroup	Study Sample	Sen [95% CI; *I*^2^, %]	Spe [95% CI; *I*^2^, %]	PLR [95% CI; *I*^2^, %]	NLR [95% CI; *I*^2^, %]	DOR [95% CI;]
Country						
China	17	0.79 [0.76, 0.82; 34.61]	0.86 * [0.81, 0.90; 82.31]	5.5 * [4.2, 7.3; 80.03]	0.24 [0.21, 0.28; 17.43]	23 * [17,31]
Korea or Turkey	6	0.80 [0.75, 0.85; 11.41]	0.77 * [0.67 0.87; 81.65]	3.6 * [2.2, 6.0; 63.86]	0.25 [0.18, 0.34;43.41]	14 * [7,30]
Study Design						
Retro	16	0.80 [0.77, 0.83; 22.27]	0.85 [0.80, 0.88; 78.68]	5.2 [3.9, 6.9; 68.84]	0.24 [0.20, 0.27; 32.55]	22 [15,32]
Pro	7	0.79 [0.73, 0.84; 45.12]	0.82 [0.68, 0.90; 90.85]	4.3 [2.5, 7.3; 79.40]	0.26 [0.21, 0.32; 0.00]	17 [10,29]
Diagnostic Basis						
AC	8	0.82 [0.77, 0.86; 11.98]	0.82 [0.77, 0.88; 81.02]	4.5 [3.1, 6.6; 60.62]	0.22 [0.18, 0.28; 0.00]	20 [13,31]
PV	2	0.81 [0.71, 0.89]	0.77 * [0.67, 0.85]	3.8 * [1.5, 9.6]	0.25 [0.15, 0.39]	16 * [5,52]
MVDP	7	0.76 [0.70, 0.80; 35.69]	0.91 * [0.88, 0.93; 0.00]	8.1 * [6.1, 10.8; 0.00]	0.27 [0.22, 0.33; 33.58]	30 * [20,46]
VI	6	0.81 [0.77, 0.85; 22.24]	0.77 [0.64, 0.87; 82.47]	3.6 [2.1, 6.0; 65.20]	0.24 [0.18, 0.32; 48.02]	15 [7,30]
Overall	23	0.80 [0.77, 0.84; 28.32]	0.84 [0.79, 0.88; 84.36]	4.9 [3.8, 6.3; 76.23]	0.24 [0.22, 0.27; 17.87]	20 [15,27]

AC, Alder classification; DOR, diagnostic odds ratio; MVDP, microvascular distribution pattern; NLR, negative likely ratio; PLR, positive likely ratio; PV, penetrating vessel; Sen, sensitivity; Spe, specificity; VI, vascular index; *: *p* < 0.05.

## Data Availability

Not applicable.

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
