# Peer review of "Diagnostic Value of Superb Microvascular Imaging in Differentiating Benign and Malignant Breast Tumors: A Systematic Review and Meta-Analysis"

_diagnostics, 2022, doi:10.3390/diagnostics12112648_

Round 1

Reviewer 1 Report

The overall impression of the technical contribution of the current study is marginal. However, the Authors may consider doing necessary amendments to the manuscript for better comprehensibility of the study.

1. The contribution of the current study must be briefly discussed as bullet points in the introduction. And motivation must also be discussed in the manuscript.
2. The overall organization of the manuscript is not discussed anywhere in the manuscript. Please add the same in the introduction section of the manuscript.
3. Figure 1 focusing on the study selection mechanism is clear, please provide a reasonable quality figure.
4. Add a minor literature section, presenting how other review studies are performed and what additional aspects the other contributed to the current review. For example, the current study is identical to  https://doi.org/10.1177/1759720X20973560
5. What are the repositories/publication houses/Quartiles considered in data acquisition?
6. From the statement "eleven studies were found to have an unclear risk of bias", what is an unclear risk of bias.? Authors must elaborate more on such sensitive statements.
7. what are groups 1,2,3 and 4 discussed in line 161, what where they and where were are groups presented.?
8.  Authors must provide the citations appropriately, for example, "Xiao et al.", the reference may be cited immediately after the author's name.
9. Authors must make it clear that "Heterogeneity Assessment" is a separate sub-section at line 148.
10. In table 2, the third row, like the country must be moved to the first row, overall may be moved to the last row.
11. Authors must provide, a citation for the sub-sub section 3.2. to make it evident.
12. From the statement "corresponding 95% confidence intervals (95% CI) [22]." what is the meaning of the 95% CI enclosed inside parenthesis?
13. Authors must present the impact analysis of the study.
14. where is the Meta-Analysis as discussed in the title summarized in the current study?
15. By considering the current form of the conclusion section, it is hard to understand by MDPI Journal readers. It should be extended with new sentences about the necessity and contributions of the study by considering the authors' opinions about the experimental results derived from some other well-known objective evaluation values if it is possible.
16. The conclusions section lacks the gap analysis and the future directions for the research and applications.
17. A thorough English proofreading is strongly recommended.

Author Response

Reviewer 1

The overall impression of the technical contribution of the current study is marginal. However, the Authors may consider doing necessary amendments to the manuscript for better comprehensibility of the study.

  1. The contribution of the current study must be briefly discussed as bullet points in the introduction. And motivation must also be discussed in the manuscript

Response: Thanks for your professional suggestion.  The contribution of our review was not clearly articulated before, therefore, we have made corresponding changes in the new version.  In summary, the basic structure of the introduction is: the statement of the importance of early detection of breast cancer → tumor neovascularization is closely associated with tumor growth, invasion, and metastasis → the advantage and limitation of CDFI and PDI in distinguishing tumor neovascularization → the advantage of SMI in characterizing tumor neovascularization → although a meta-analysis addressed this topic was published,it has some pitfalls and it may not reveal the diagnostic accuracy of SMI in differentiating benign and malignant breast tumors. Moreover, there are many diagnostic criteria for SMI to diagnose breast tumors,the choice of diagnostic basis is a key challenge for sonographers in clinical practice. Hence, the primary goal of our review is to evaluate the diagnostic value of SMI in differentiating benign and malignant breast tumors, what’s more, we also conducted subgroup analysis to investigate which diagnostic criteria has the best diagnostic efficiency. We sincerely hope that our work can be helpful to clinical practice.

  1. The overall organization of the manuscript is not discussed anywhere in the manuscript. Please add the same in the introduction section of the manuscript.

Response: We are very grateful for your professional advice. We have added overall organization of the manuscript in the introduction section (line 125-128).

  1. Figure 1 focusing on the study selection mechanism is clear, please provide a reasonable quality figure.

Response: I'm sorry for the inconvenience caused by the unclear picture, we have now re-uploaded the picture in the manuscript, and the quality of the picture meets the requirements of MDPI.

  1. Add a minor literature section, presenting how other review studies are performed and what additional aspects the other contributed to the current review. For example, the current study is identical to https://doi.org/10.1177/1759720X20973560

Response:  Thank you for your suggestion. We really appreciate the article you gave with examples and it has benefited us a lot, We have presented it in the Method section(lines 133-134)

  1. What are the repositories/publication houses/Quartiles considered in data acquisition?

Response: We are very grateful for your professional suggestion. In our study, the EndNote software 20.0 was used to manage the literature retrieved from electronic database (PubMed, Embase, Cochran Library, Web of Science, China Biology Medicine Disc, China National Knowledge Infrastructure, Wanfang, and Vip databases). Moreover, we screened the literature according to inclusion and exclusion criteria. The data were independently extracted by 2 reviewers and stored in Microsoft Excel spreadsheets, what’s more, except for the baseline data, the main data we extracted were the TP, FP, FN, and TN. As for publication houses consideration, We have carried out a relatively comprehensive literature search in the database. As long as the articles meet our inclusion and ranking criteria, we will download the full text to minimize the publication bias caused by ignoring the articles of some publication houses. However, this was not very clear in our previous manuscript. We have added elaboration in the new version of manuscript (Lines 137-139, lines 150-151, lines 167-168).

  1. From the statement "eleven studies were found to have an unclear risk of bias", what is an unclear risk of bias.? Authors must elaborate more on such sensitive statements.

Response: We really appreciate your professional advice. It is very necessary to elaborate more on such sensitive statements. We have added these elaborations in section 2.4 Quality Assessment (lines 181-184).

  1. what are groups 1,2,3 and 4 discussed in line 161, what where they and where were groups presented.?

Response: Thanks for your professional suggestion. The reason why we made such a grouping in our study are briefly explained as follows:at present there are four main diagnostic criteria for SMI to diagnose breast tumors, namely: AC, PV, MVDP and VI. However, their diagnostic accuracy varies, which can be confusing for sonographers as to which diagnostic criteria to choose. Hence, we conducted subgroup analysis to investigate which diagnostic criteria has the best diagnostic performance. We have presented these results in Table 2 and supplementary material (lines 351-354).

  1. Authors must provide the citations appropriately, for example, "Xiao et al.", the reference may be cited immediately after the author's name.

Response: Thank you for your suggestion. We have cited references immediately after all author names in the manuscript. Thank you very much.

  1. Authors must make it clear that "Heterogeneity Assessment" is a separate sub-section at line 148.

Response: We really appreciate your valuable suggestions. The sub-section of "Heterogeneity Assessment" should have been included in the "Quantitative synthesis”. But we wanted to show more details of our grouping so that the source of the heterogeneity could be more easily understood by the readers, therefore, we conducted it as a separate sub-section. 

  1. In table 2, the third row, like the country must be moved to the first row, overall may be moved to the last row.

Response: Thanks for your insightful comment. We have made the corresponding changes (lines 351-354).

  1. Authors must provide, a citation for the sub-sub section 3.2. to make it evident.

Response: Thanks for your professional advice. We agree that all viewpoint should be supported by relevant references. In the revised manuscript, we have cited the references (lines 237-239)

  1. From the statement "corresponding 95% confidence intervals (95% CI) [22]." what is the meaning of the 95% CI enclosed inside parenthesis?

Response: Thank you for pointing out this error. We apologize for the carelessness, in addition, the original meaning of the 95% CI enclosed inside parenthesis is the abbreviation of the previous term, and its correct form should be corresponding 95% confidence intervals (CI). Thanks a lot for your correction.

  1. Authors must present the impact analysis of the study.

Response: Thanks for your professional and insightful comment. We have presented the impact analysis on lines 439-483.

  1. where is the Meta-Analysis as discussed in the title summarized in the current study?

Response: We really appreciate for your professional suggestion. The organization of our previous manuscript was not very clear, which might make it difficult for readers to find the section of “Meta-Analysis”. Hence, we studied the articles you suggested. Then we modified our article organization (lines 187-219 and lines 284-349). 

  1. By considering the current form of the conclusion section, it is hard to understand by MDPI Journal readers. It should be extended with new sentences about the necessity and contributions of the study by considering the authors' opinions about the experimental results derived from some other well-known objective evaluation values if it is possible.

Response: Thank you for your professional advice. To be honest, the statement of the necessity and contributions of our work was not expressed clearly in the previous version of our manuscript. Hence, we extended with new sentences including but not limited to conclusion section (lines 90-98, lines 452-453, lines 481-483 and lines 350-353). However, SMI is a novel doppler technology arising in the recent year, maybe it is difficult to find well-known objective evaluation values of this topic.

  1. The conclusions section lacks the gap analysis and the future directions for the research and applications.

Response: We really appreciate for your insightful comment. We have added the gap analysis of our study and the future prospects of application of SMI in differentiating breast lesions (lines 485-494).

  1. A thorough English proofreading is strongly recommended.

Response: Thanks for your suggestion. The revised manuscript has been edited and proofread for all language-related errors by Medjaden Biosciences Inc., a medical editing company. We hope this revision can meet your requirements. Last but not least, if there is anything that does not meet the requirements, please be not hesitate to give your insightful comments, we will try our best to revise the article.

Reviewer 2 Report

·         The novelty of this paper is not clear. The difference between present work and previous Works should be highlighted.

·         The author needs to change the abstract and focus more on problem domain. Before the paper contributions, the author could precisely include the need of developing the proposed method.

·         Are there other factors to compare modern methods, such as time execution?   Other meaning  how did the authors apply the Augmentation technique?

·         The author could better explain how “Related works” is actually related to the current study. It is not clear to the reader how the manuscript is similar to or differs from these related works.

·         Some recent works should be added, such as:  https://doi.org/10.3390/biology11030439  

·         The manuscript is well-organized and properly formatted. The authors are suggested to have the paper revised to improve the language.

·         Is there any limitation of the proposed work? If so, the author should include it at the end of conclusion part. This may help future researchers to overcome the limitations.

Author Response

1· The author needs to change the abstract and focus more on problem domain. Before the paper contributions, the author could precisely include the need of developing the proposed method.

Response: We appreciate for your insightful suggestion. We have revised the structure of the article in the new version to make readers more aware of the necessity and contribution of our research. the basic structure of the introduction is: the statement of the importance of early detection of breast cancer → tumor neovascularization is closely associated with tumor growth, invasion, and metastasis → the advantage and limitation of CDFI and PDI in distinguishing tumor neovascularization → the advantage of SMI in characterizing tumor neovascularization → although a meta-analysis addressed this topic was published,it has some pitfalls and it may not reveal the diagnostic accuracy of SMI in differentiating benign and malignant breast tumors. This is the motivation and necessity of our systematic review.

2· Are there other factors to compare modern methods, such as time execution?   Other meaning how did the authors apply the Augmentation technique?

Response: We appreciate for your professional advice. After receiving your comment, we carefully read some of the high-quality systematic review and make a considerable change of our “Materials and Methods” and “Results” section. However, we must be sorry that we didn't quite understand what the meaning of time execution. Can I trouble you to be more specific? If time execution is referred to time consumption of SMI examination, it may spend 10-15 second to obtain cine clips of the richest vasculature planes (sagittal and transverse planes) of breast lesions. Therefore, the time consumption of the SMI examination is much shorter than that of the CEUS examination. However, SMI is equally effective at identifying breast lesions as CEUS, hence, SMI is promising tool for the diagnosis of breast tumor. In addition, if the meaning of Augmentation technique is SMI, grayscale ultrasound is first performed to scan the breast lesions thoroughly, SMI is then performed to evaluate the vascularity in and around the lesions and two orthogonal planes were obtained for each lesion with the richest vasculature. you may can review its application in clinical practice on the lines 439-483.

3· The author could better explain how “Related works” is actually related to the current study. It is not clear to the reader how the manuscript is similar to or differs from these related works.

Response: We appreciate for your professional advice. We have added a sub-section named “Comparison with Previous Systematic Review” (lines 426-438).

4· Some recent works should be added, such as:  https://doi.org/10.3390/biology11030439  

Response: We are very grateful for your careful suggestion. The emergence of a novel Doppler technology, such as SMI, is inseparable from the rapid development of computer technology. We cannot agree more with it. Thanks for your professional advice, we have added some resent works to our “Introduction” section. You can review it on the lines 71.

 5· The manuscript is well-organized and properly formatted. The authors are suggested to have the paper revised to improve the language.

Response: I appreciate your compliments on my part of the discussion, but it still leaves something to be desired. As for language improvement, the revised manuscript has been edited and proofread for all language-related errors by Medjaden Biosciences Inc., a professional medical editing company. We hope this revision can meet your requirements.

6· Is there any limitation of the proposed work? If so, the author should include it at the end of conclusion part. This may help future researchers to overcome the limitations

Response: We very appreciate for your insightful comment. We have presented the limitation of our proposed work in the sub-section: “Limitations of our Work”. You can review it on lines 485-493.

In addition, we also summarized it in the section: “Conclusion”, you can review it on lines 526-528. All in all, we really appreciate for your professional and insightful suggestion. If there is anything that does not meet the requirements, please be not hesitate to give your insightful comments, we will do our best to revise the article.

Round 2

Reviewer 1 Report

The authors have addressed the recommendations in a reasonable manner. Manuscript in their current form may be processed to the next phase of the editorial process.

Authors may check for typos and grammar mistakes.

And also check if the conclusion is adequately presented.

Author Response

The authors have addressed the recommendations in a reasonable manner. Manuscript in their current form may be processed to the next phase of the editorial process.

  1. Authors may check for typos and grammar mistakes.

Response:We very appreciate for your professional comment. As for language improvement, the revised manuscript has been edited and proofread by Medjaden Biosciences Inc again. We hope this revision can meet your requirements.

  1. And also check if the conclusion is adequately presented.

Response: We are very grateful for your professional suggestion. We have extended with new sentences in the new version of manuscript (Lines 382-390). 

Reviewer 2 Report

The paper becomes good 

Author Response

The paper becomes good 

Response: First of all, we appreciate your compliments on our manuscript. More importantly, the improvement of manuscript quality is inseparable from your meticulous review work, so we thank you again for your effort.